# Differences in lower extremity kinematics during single-leg lateral drop landing of healthy individuals, injured but asymptomatic patients, and patients with chronic ankle instability- a cross-sectional observational study

Yuki Sagawa[1]*, Takumi Yamada[2], Takehiro Ohmi[3], Yoshinao Moriyama[4], Junpei Kato[5]

1 Department of Rehabilitation, Sonodakai Joint Replacement Center Hospital, Tokyo, Japan, 2 Department of Physical Therapy, Faculty of Health Sciences, Tokyo Metropolitan University, Tokyo, Japan, 3 Clinical Center for Sports Medicine and Sports Dentistry, Tokyo Medical and Dental University, Tokyo, Japan, 4 Department of Rehabilitation, Division of Physical Therapy, Tokyo Metropolitan Rehabilitation Hospital, Tokyo, Japan, 5 Karadacare Business Development Office, NEC Livex, Ltd., Tokyo, Japan

* sagawapt@yahoo.co.jp

## Abstract

The lower-extremity kinematics associated with forward jump landing after an ankle injury is known to differ for patients with Chronic Ankle Instability (CAI), copers (injured but asymptomatic patients), and healthy individuals. However, the differences in the lower extremity kinematics of these groups associated with a Single-leg Lateral Drop Landing (SLDL) are unknown. The purpose of this study is to characterize the lower limb and foot kinematics during SLDL in CAI patients and to compare these characteristics with those of the copers and healthy individuals. This was a cross-sectional observational study. Nineteen participants, each, were selected from the CAI, Coper, and control groups. The lower-extremity kinematics during SLDL was measured using three-dimensional motion analysis over an interval progressing from 200 ms before landing to 200 ms after landing. Either one-way ANOVA or the Kruskal–Wallis test was used to compare the attributes of the respective groups, with each parameter measured every 10 ms. The maximum values and excursions of the parameters were established over time intervals progressing from 200 ms before landing to 200 ms after landing. Significant observations were subjected to post hoc analysis. Compared to the Coper group, the CAI group exhibited significantly smaller hip adduction angles at 160 ms, ankle dorsiflexion angles in the 110–150 ms interval, and maximum ankle dorsiflexion angles after landing. Compared to the control group, the CAI group exhibited significantly smaller excursions of MH inversion/eversion after landing. Our findings confirm the necessity of focusing on the kinematics of hip adduction/abduction and plantar/dorsiflexion during SLDL in evaluating patients with ankle injuries.

**Data Availability Statement:** All relevant data are within the manuscript and its Supporting information files.

**Funding:** Initials of the authors who received each award: Yuki Sagawa Grant numbers awarded to each author: 2021002 The full name of each funder: Tokyo Physical Therapists Association URL of each funder website: http://www.pttokyo.net/topics/2021/04/15476.html Did the sponsors or funders play any role in the study design, data collection and analysis, decision to publish, or preparation of the manuscript?: Gratuities to subjects.

**Competing interests:** The authors have declared that no competing interests exist.

## Introduction

Ankle sprains are the most frequent form of sports injury, accounting for 15–21% of all sports injuries [1, 2]. Approximately 80% of these ankle sprains are Lateral Ankle Sprains (LASs) [2]. Forty percent of LAS injuries progress to Chronic Ankle Instability (CAI) [2]. CAI has been recognized as a possible cause of other sports-related disorders [3] that impact sports activities [4]. Therefore, it is necessary to prevent progression from LAS to CAI.

CAI symptoms include pathological joint laxity, ankle joint pain, and instability [5]. These CAI symptoms appear during movements such as jump landings [6–9] and persist for an extended period [4]. Patients with a history of LAS but without the symptoms of CAI (copers) have been observed to function without pain or ankle instability [10]. We may infer that the biomechanics of movement could differ between CAI patients, copers, and healthy individuals.

Several studies have been conducted on the kinematics of the lower limbs during movement in patients with CAI. Compared to copers and healthy individuals, CAI patients exhibit smaller ankle dorsiflexion angles [11–14] and ankle plantar/dorsiflexion excursions [15] during jogging and after jump landings. There are reports that there are no group differences in subtalar inversion/eversion or adduction/abduction angles during jump landing movements among CAI patients, copers, and healthy individuals [11, 12]. Compared to healthy individuals, CAI patients and copers exhibit larger inversions of the joints between the metatarsals and the hindfoot (the MH joint) during side jumps [16], but these observations are based on a small number of studies. As described earlier, the differences in lower limb kinematics between CAI patients, copers, or healthy individuals may not be limited to ankles alone.

Movements that occur primarily in the frontal plane cause tensile stress on the eversion muscle of the ankle and the lateral ligaments of the ankle joint [6, 17], which may influence the occurrence of LAS and the progression of CAI. LAS is often associated with jump landings [1, 8, 9], and sports that require many jump landings also involve many lateral jump landings [18]. Therefore, understanding the biomechanics of Single-leg Lateral Drop Landing (SLDL) is important for preventing LAS and the transition to CAI. Previous studies have regarded forward jump landings [13, 19, 20] and side jumps [16, 20] as motor tasks; no previous studies have examined lower-extremity kinematics during SLDL among the three groups of CAI patients described above. The study objective was to compare the characteristics of hip, knee, ankle, subtalar, MH, and the joints between the toes and the metatarsals (TM) angles during SLDL that may be involved in the occurrence of LAS and the transition to CAI, by differentiating between such characteristics in CAI patients, copers, and healthy individuals. We hypothesized that CAI patients would have smaller ankle dorsiflexion angles and larger MH inversion angles after landing, compared to copers and healthy individuals, and tested the validity of this hypothesis.

## Materials and methods

### Design

We conducted a cross-sectional observational study across three groups: CAI patients, copers, and healthy individuals (control). The difference in movement kinematics between the three groups made it necessary to select a design using three groups instead of two.

### Sample size estimation

The sample size was calculated for an effect size of 0.40, an $\alpha$ value of 0.05, and 80% power based on prior studies [21]. The G*Power 3.1.9.7. statistical power analysis tool was used. The

analysis in this study was set up with twenty-two CAI group members (twenty-two legs), twenty-two Coper group members (twenty-two legs), and twenty-two CON group members (twenty-two legs), for a total of sixty-six participants and sixty-six legs [21].

## Sampling technique

Participants were selected by convenience sampling.

## Participants

The inclusion criteria follow. Participants were recruited for this study between February 25, 2021 and November 30, 2021. They were males aged 18–30 years [22], with a Tegner Activity level of ≧3 [23]. The study participants were divided into three groups: those with CAI (CAI group), those with a history of LAS but without pain or instability (Coper group), and those without a history of LAS (CON (control) group).

The inclusion criteria for the CAI group were set as follows: (1) at least one incidence of LAS with pain or loss of function on the side examined [21], (2) a history of giving way [24, 25], and (3) a score of < 25 points in the Japanese version of the Cumberland Ankle Instability Tool (CAIT) [26]. The CAI group was defined to include participants with a history of LAS in both legs; the leg to be measured was the leg with the lower Japanese CAIT score [27].

The inclusion criteria for the Coper group were set as follows: (1) at least one incidence of LAS on the side examined [10, 21], (2) no history of giving way, ankle instability, or recurrence of LAS within 1 year [10], (3) a score of ≧ 25 points in the Japanese CAIT, and (4) no CAI on the non-examined side. In the Coper group, for participants with a history of LAS in both legs, the leg with the higher Japanese CAIT score was selected for measurement.

The inclusion criteria for the CON group were set as follows: (1) no history of LAS on the side examined [21], (2) no CAI on the non-examined side, and (3) a score of 30 points in the Japanese CAIT. For the CON group, the dominant leg was used for measurement, as there was no history of LAS in either leg.

Exclusion criteria were established with reference to previous studies [19, 21, 25], as follows: (1) participants with a history of lower extremity surgery, (2) participants who had an acute lower extremity injury, within three months of study participation, that limited the desired physical activity for a day, (3) participants with neurological disease, vestibular impairment, or a visual impairment, and (4) participants who were fearful of the motor tasks in the study.

A flow diagram of participant recruitment is shown in Fig 1.

## Setting

Prior to the measurement, all participants were instructed to be barefoot, shirtless, and to wear spandex shorts in their underclothes. Markers with a diameter of 9 mm were applied to 67 locations on each subject's entire body, following the Plug-in Gait Full-Body model (https://www.vicon.com/) (Fig 2) for the upper body, thighs and lower legs, and the multi-segment foot model method [28, 29] (Fig 3) for the feet. Motion analysis was performed using a three-dimensional motion analysis system (Vicon Nexus, Oxford, London, UK) with twelve infrared cameras with a sampling frequency of 100 Hz. The ground reaction forces were recorded using a dual force plate (Kisler Japan, Tokyo, Japan) with a sampling frequency of 1,000 Hz. A software for interactive musculoskeletal modeling (SIMM; MusculoGraphics, Santa Rosa, CA, USA) was used to analyze the kinematics of the foot and lower extremity based on three-dimensional data.

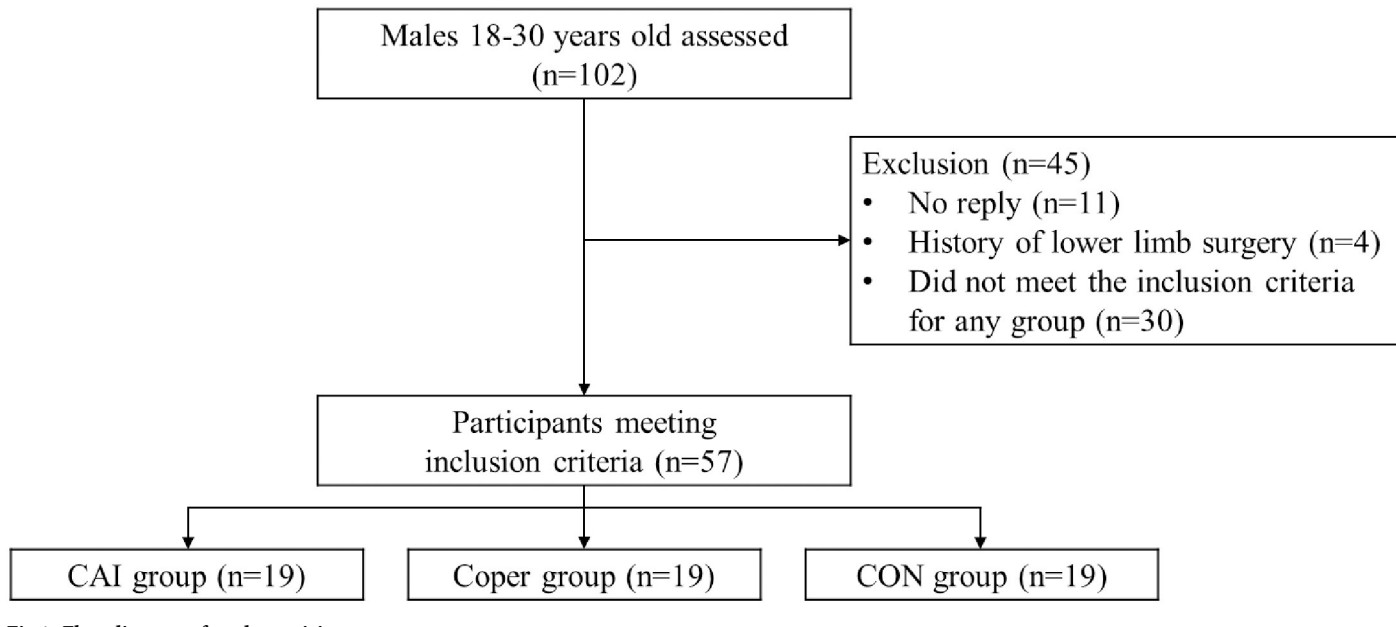

**Fig 1. Flow diagram of study participants.**

## Procedure

The marker coordinates were measured while standing stationary on the force plate. SLDL was set up as a motor task [30]. The starting position was a two-legged standing position on a 20 cm platform (Fig 4), based on earlier study [30]. The subject was positioned one-legged with both upper extremities crossed in front of the chest, with the test side as the supporting side (Fig 4). The test subject then landed on the center of the force plate, 30 cm away, on the test side only (Fig 4). The jump height at this time was the minimum required. After landing on the plate, the posture was held for 5 s (Fig 4) [20]. Participants were allowed to practice three times and then the motor task was performed until three successful attempts had been made [30]. Based on previous reports, failure of the motor task was defined as meeting any

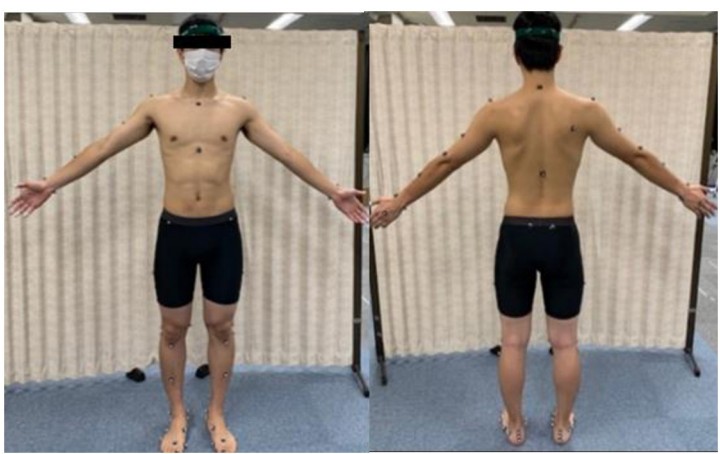

**Fig 2. Marker attachment positions according to the Plug-in Gait Full-Body model.**

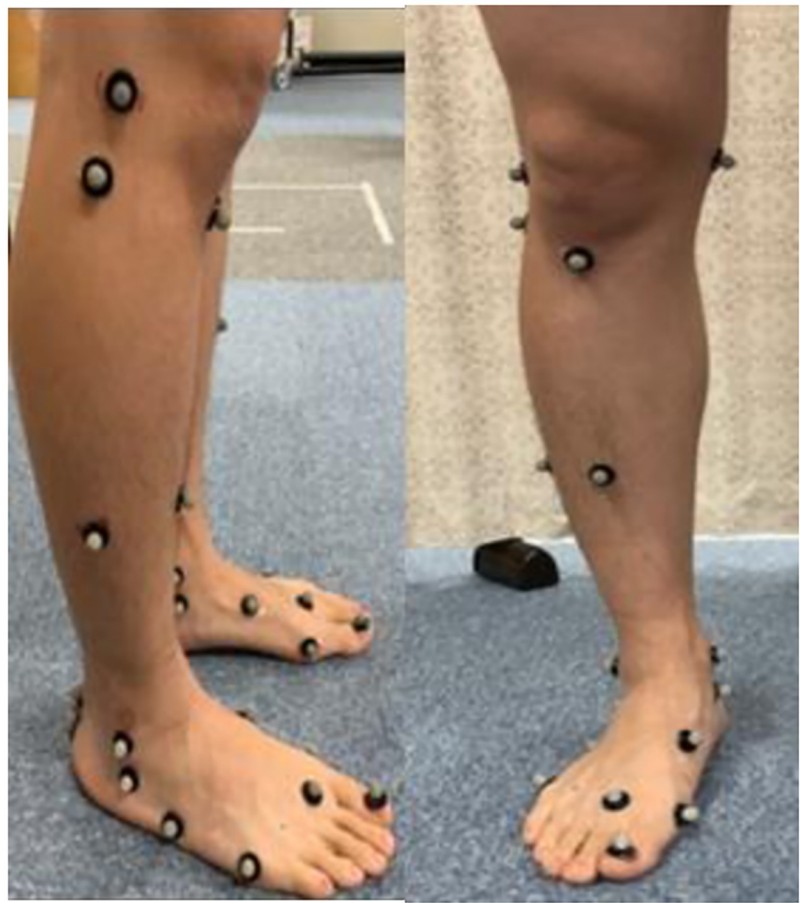

**Fig 3. Marker attachment positions according to the multi-segment foot model.**

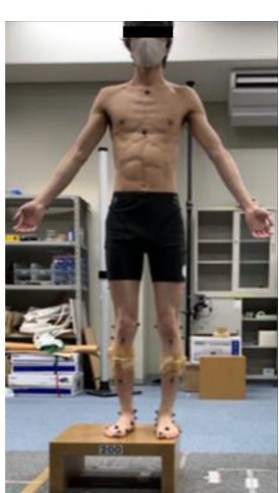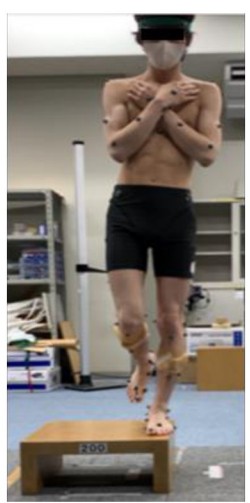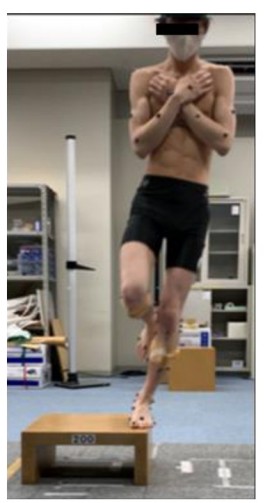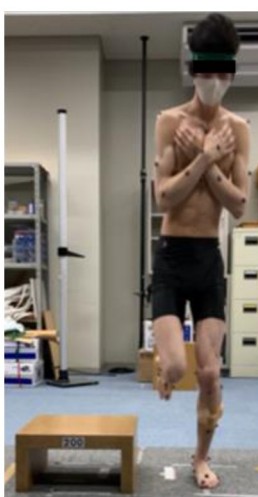

**Fig 4. Single-Leg Lateral Drop Landing (SLDL).**

one of the following criteria [30]: (1) landing partially off the force plate, (2) foot movement or sliding after landing, (3) the opposite foot touching the force plate or floor, (4) excessive hip or trunk deviation. Failure of the motor task was determined visually and no feedback on the landing was given [30].

### Data processing

The missing parts of the marker trajectory data were completed by the gap-filling function in Vicon. Marker trajectory data and ground reaction force data were smoothed using the Butter worth filter in Vicon, with reference to a previous study [29].

The ground reaction force data and three-dimensional data of static standing and SLDL obtained by the above method were imported into the SIMM to create musculoskeletal models of the entire body and foot [29]. The degrees of freedom for each segment were as follows: hip joint–three, knee joint–one, ankle joint–one, subtalar joint–two, MH joint–three, and TM joint–one. The angles of hip flexion/extension, adduction/abduction, internal/external rotation, knee flexion/extension, plantar/dorsiflexion, subtalar adduction/abduction, inversion/eversion, MH plantar/dorsiflexion, adduction/abduction, inversion/eversion, and TM flexion/extension in the 200 ms interval before and after landing were calculated using the musculoskeletal model created in the SIMM. The angles of hip flexion, adduction and internal rotation, knee flexion, ankle dorsiflexion, subtalar adduction and inversion, MH dorsiflexion, adduction and inversion, and TM extension were expressed as positive values. The average of three trials was used for the analysis. The definition of initial contact was set when the vertical component of the ground reaction force exceeded 10 N [30]. For the analysis section, the initial contact point was set at 0 ms, and measurements were taken every 10 ms from -200 ms to 200 ms during the motor task. These data were anonymized, and no individual participant information was accessed during or after data collection.

### Primary and secondary outcome variables

The primary outcome was set for ankle and subtalar kinematics, and the secondary outcome was set for hip, knee, MH, and TM kinematics.

### Statistical analysis

The Shapiro-Wilk test was used to confirm the normality of each parameter. Either one-way ANOVA or the Kruskal–Wallis test was used to compare the attribute data among the groups, measuring each parameter every 10 ms. The maximum value and excursion of each parameter were established over the time interval progressing from 200 ms before landing to 200 ms after landing. Either the Tukey or Steel-Dwass test was used in the post hoc analysis of significant parameters. Easy R (EZR) [31] was used for all statistical analyses and the significance level was set to 5%. The effect size $\eta^2$ was calculated to determine the magnitudes of the differences in the dependent variables between the groups. The criteria for determining the size of the effect were set to 'small' ($\eta^2 < 0.06$), 'moderate' ($0.06 \leq \eta^2 < 0.14$), and 'large' ($0.14 \leq \eta^2$) [32].

### Ethics

The study protocol was approved by the Tokyo Research Safety Ethics Committee of the Tokyo Metropolitan University (Approval No. 20053). The study participants were fully informed of the content and purpose of the research, and the study procedures were carried out after obtaining written informed consent from the participants.

**Table 1. Participant demographic data.**

|  | CAI (n = 19) | Coper (n = 19) | CON (n = 19) | P-value | Effect Size |
|---|---|---|---|---|---|
| Age (years) | 25.3 ± 2.9 | 24.6 ± 2.5 | 23.9 ± 2.8 | .282 | .05 |
| Height (cm) | 174.3 ± 5.7 | 171.9 ± 4.4 | 171.4 ± 7.1 | .282 | .05 |
| Body weight (kg) | 70.9 ± 10.8 | 63.5 ± 8.3 | 63.4 ± 10.3 | .055 | .12 |
| CAIT (point)* | 22 (4.0) | 28 (1.0) | 30 (0.0) | < .01 | .75 |

Abbreviations: CAI, Chronic Ankle Instability; CON, control; CAIT, Cumberland Ankle Instability Tool

Mean ± SD; Median (interquartile range)

*CAI group, Coper group and CON group differ significantly at p < .05

## Results

A total of 57 patients were included in the analysis: 19 in the CAI group, 19 in the Coper group, and 19 in the CON group. Basic demographic data for each group are presented in Table 1.

### Ankle/subtalar angle

Compared to the Coper group, the CAI group exhibited significantly smaller ankle dorsiflexion angles in the 110–150 ms interval (Fig 5), and maximum ankle dorsiflexion angles after landing (Table 3). Other outcomes were not significantly different (Figs 6 and 7, Tables 2 and 3).

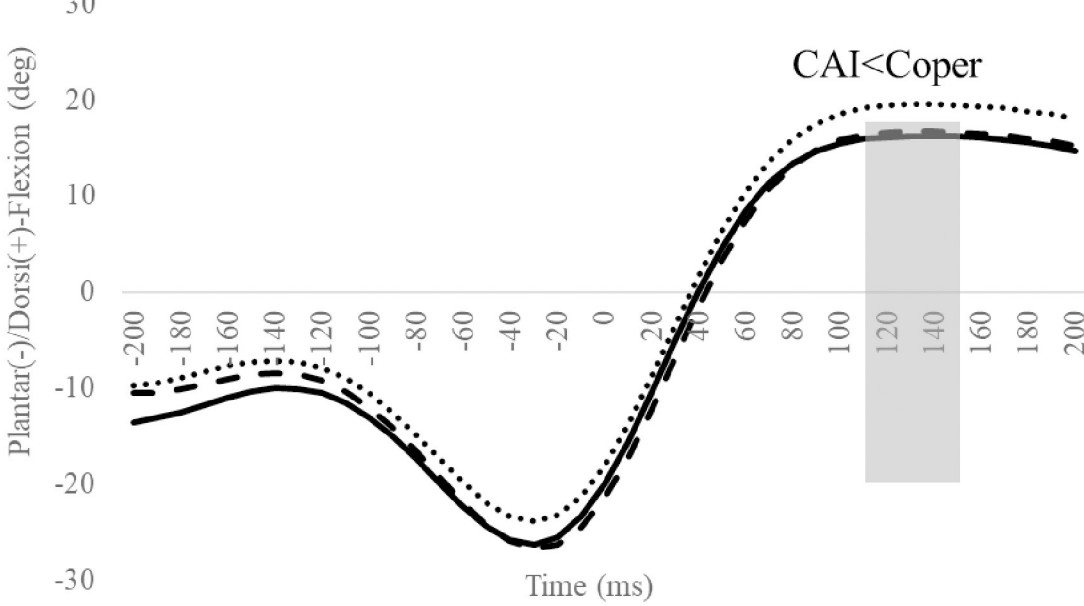

**Fig 5. Sagittal ankle angles during SLDL from 200 ms before landing to 200 ms after landing.** The continuous, dotted, and dashed lines represent CAI, Coper, and CON, respectively. The time origin at 0 ms indicates the moment of initial contact. Gray areas indicate areas of statistical significance. Abbreviations: SLDL, Single-leg Lateral Drop Landing; CAI, Chronic Ankle Instability; CON, control.

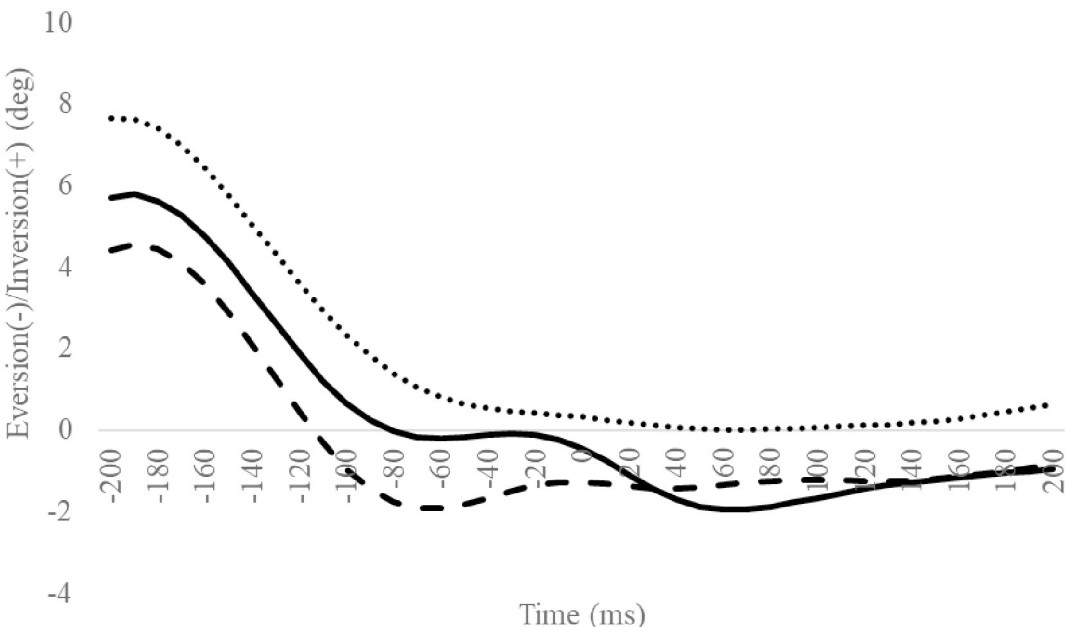

**Fig 6. Frontal subtalar angles during SLDL from 200 ms before landing to 200 ms after landing.** The continuous, dotted, and dashed lines represent CAI, Coper, and CON, respectively. The time origin at 0 ms indicates the moment of initial contact. Gray areas indicate areas of statistical significance. Abbreviations: SLDL, Single-leg Lateral Drop Landing; CAI, Chronic Ankle Instability; CON, control.

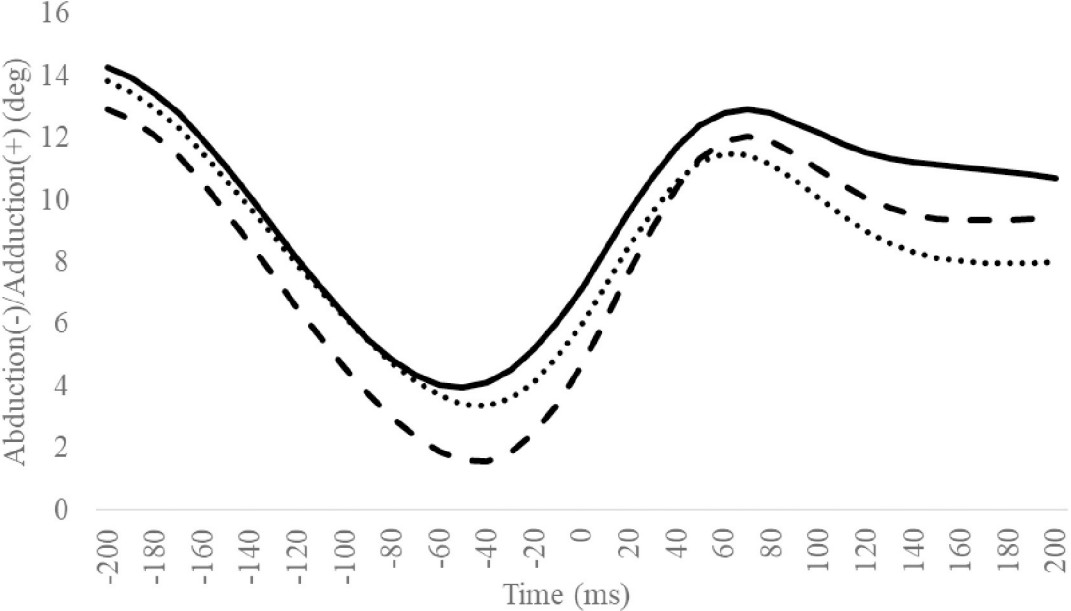

**Fig 7. Horizontal subtalar angles during SLDL from 200 ms before landing to 200 ms after landing.** The continuous, dotted, and dashed lines represent CAI, Coper, and CON, respectively. The time origin at 0 ms indicates the moment of initial contact. Gray areas indicate areas of statistical significance. Abbreviations: SLDL, Single-leg Lateral Drop Landing; CAI, Chronic Ankle Instability; CON, control.

**Table 2. Ankle/subtalar kinematics data during 200 ms interval pre-landing.**

| | CAI | Coper | CON | P-value | Effect Size |
|---|---|---|---|---|---|
| Ankle Plantarflexion maximum (deg) | 26.9 (6.2) | 26.6 (14.5) | 27.3 (4.2) | .738 | .04 |
| Ankle Dorsiflexion maximum (deg) | −9.3 ± 6.1 | −6.3 ± 6.6 | −7.9 ± 4.1 | .259 | .05 |
| Sagittal plane excursion (deg) | 17.1 ± 4.1 | 18.0 ± 5.8 | 19.0 ± 3.4 | .316 | .03 |
| Subtalar Inversion maximum (deg) | 6.4 ± 6.8 | 8.1 ± 6.6 | 4.9 ± 6.6 | .343 | .04 |
| Subtalar Eversion maximum (deg) | 1.1 ± 5.3 | 0.5 ± 5.0 | 2.5 ± 5.6 | .497 | .03 |
| Frontal plane excursion (deg) | 7.5 ± 2.9 | 8.6 ± 3.9 | 7.4 ± 2.6 | .475 | .03 |
| Subtalar Adduction maximum (deg) | 14.3 ± 4.5 | 14.0 ± 4.8 | 13.0 ± 5.1 | .682 | .01 |
| Subtalar Abduction maximum (deg) | −3.6 ± 4.5 | −3.2 ± 3.5 | −1.4 ± 3.9 | .213 | .06 |
| Horizonal plane excursion (deg) | 10.7 ± 3.2 | 10.8 ± 3.1 | 11.6 ± 2.5 | .606 | .02 |

Abbreviations: CAI, Chronic Ankle Instability; CON, control.

Mean ± SD; Median (interquartile range)

## Hip angle

Compared to the Coper group, the CAI group exhibited significantly smaller hip adduction angles at 160 ms (Fig 8). Other hip angle outcomes were not significantly different (Figs 9 and 10, Tables 4 and 5).

## Knee angle

For all outcomes, the CAI group showed no significant differences (Fig 11, Tables 6 and 7) when compared to the Coper and CON groups.

## TM/MH angle

Compared to the CON group, the CAI group exhibited a significantly smaller excursion of the MH inversion/eversion (Table 9). The other outcomes were not significantly different (Figs 12–15, Tables 8 and 9).

**Table 3. Ankle/subtalar kinematics data during 200 ms interval post-landing.**

| | CAI | Coper | CON | P-value | Effect Size |
|---|---|---|---|---|---|
| Ankle Plantarflexion maximum (deg) | 20.0 ± 6.5 | 18.1 ± 7.8 | 21.4 ± 4.1 | .245 | .05 |
| Ankle Dorsiflexion maximum (deg)* | 16.6 ± 4.1 | 20.0 ± 3.5 | 17.1 ± 4.7 | .027 | .13 |
| Sagittal plane excursion (deg) | 36.6 ± 6.2 | 38.1 ± 7.5 | 38.5 ± 5.0 | .617 | .02 |
| Subtalar Inversion maximum (deg) | 1.0 (6.1) | 1.0 (7.3) | −0.4 (8.9) | .652 | .01 |
| Subtalar Eversion maximum (deg) | 3.2 ± 6.2 | 1.4 ± 6.3 | 3.7 ± 7.3 | .517 | .02 |
| Frontal plane excursion (deg) | 3.3 (3.9) | 2.5 (2.6) | 4.5 (3.3) | .075 | .07 |
| Subtalar Adduction maximum (deg) | 13.9 ± 4.9 | 12.0 ± 4.0 | 12.5 ± 5.9 | .484 | .03 |
| Subtalar Abduction maximum (deg) | −6.2 ± 4.2 | −5.1 ± 4.1 | −4.5 ± 4.1 | .448 | .03 |
| Horizonal plane excursion (deg) | 7.4 (6.6) | 6.0 (2.9) | 7.3 (4.8) | .600 | .02 |

Abbreviations: CAI, Chronic Ankle Instability; CON, control.

Mean ± SD; Median (interquartile range)

*CAI group and Coper group differ significantly at $p < .05$

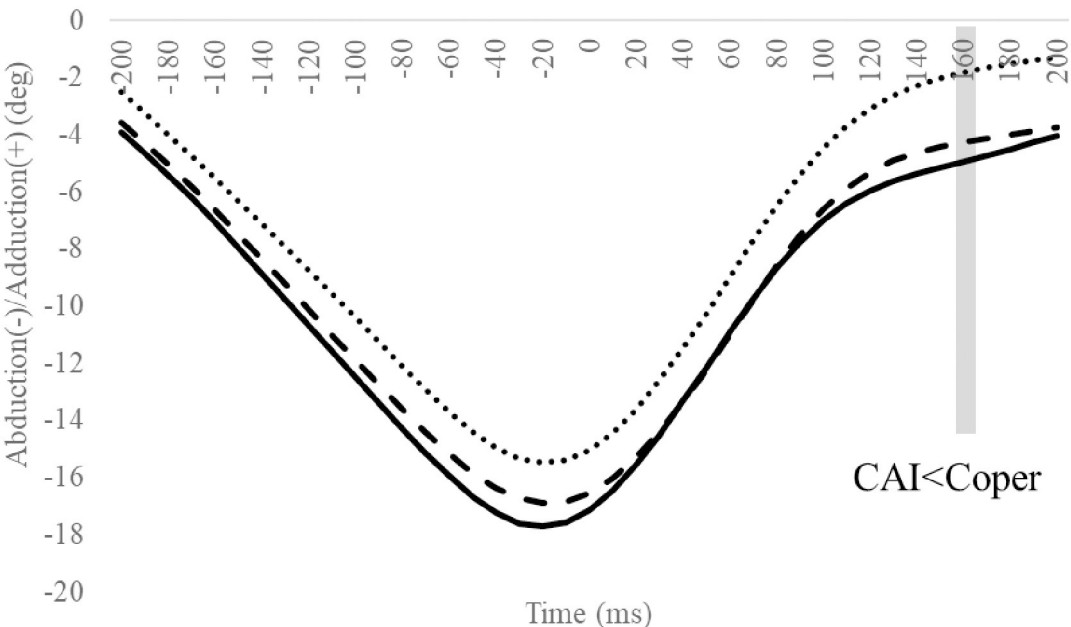

**Fig 8. Frontal hip angle during SLDL from 200 ms before landing to 200 ms after landing.** The continuous, dotted, and dashed lines represent CAI, Coper, and CON, respectively. The time origin at 0 ms indicates the moment of initial contact. Gray areas indicate areas of statistical significance. Abbreviations: SLDL, Single-leg Lateral Drop Landing; CAI, Chronic Ankle Instability; CON, control.

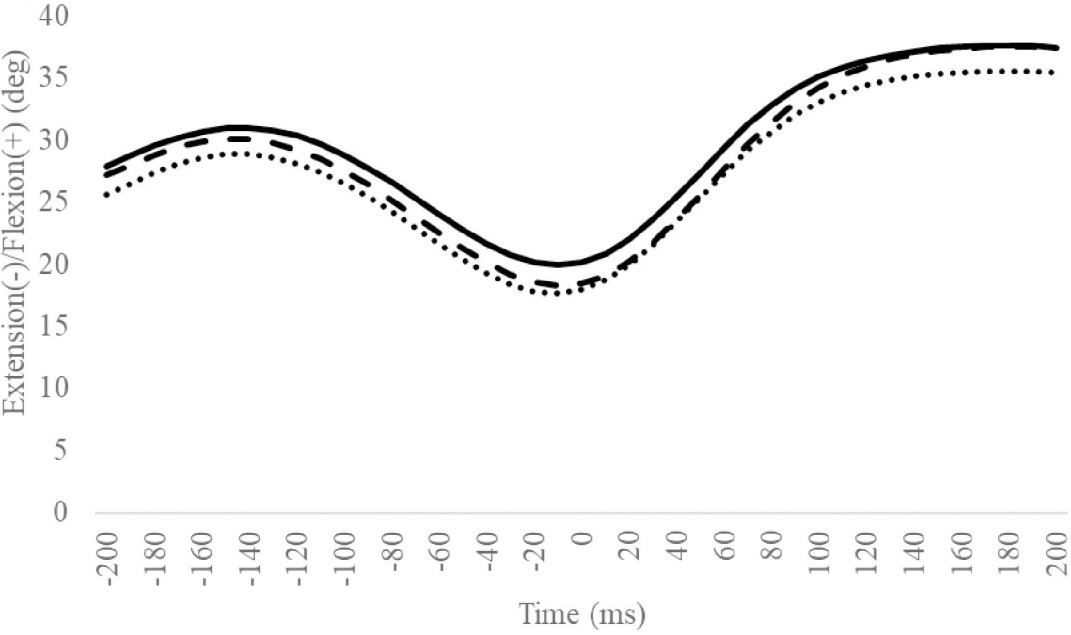

**Fig 9. Sagittal hip angle during SLDL from 200 ms before landing to 200 ms after landing.** The continuous, dotted, and dashed lines represent CAI, Coper, and CON, respectively. The time origin at 0 ms indicates the moment of initial contact. Gray areas indicate areas of statistical significance. Abbreviations: SLDL, Single-leg Lateral Drop Landing; CAI, Chronic Ankle Instability; CON, control.

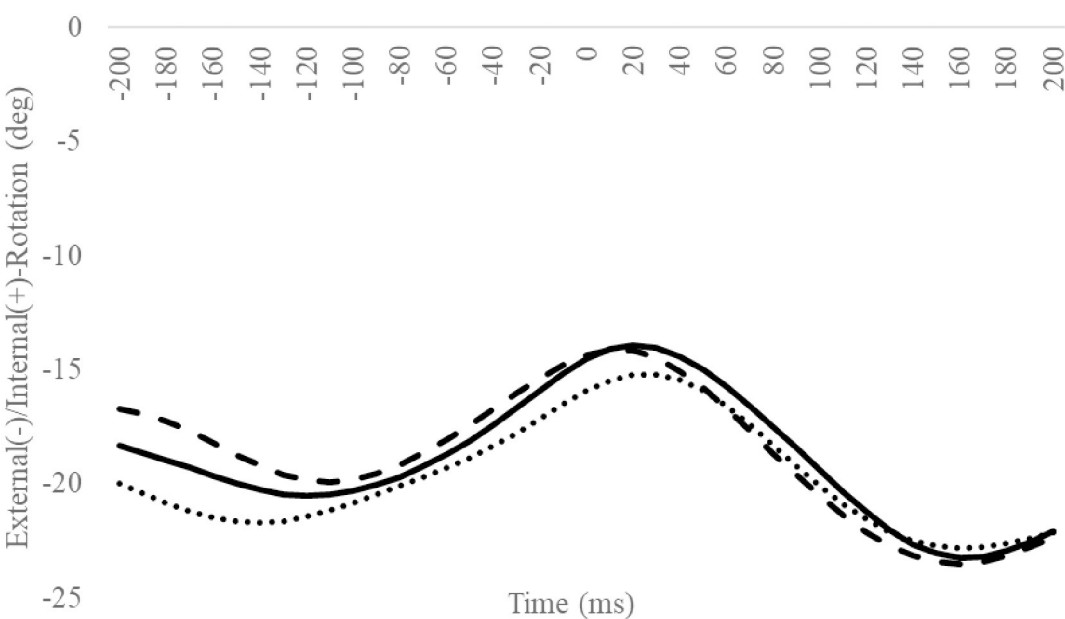

**Fig 10. Horizontal hip angle during SLDL from 200 ms before landing to 200 ms after landing.** The continuous, dotted, and dashed lines represent CAI, Coper, and CON, respectively. The time origin at 0 ms indicates the moment of initial contact. Gray areas indicate areas of statistical significance. Abbreviations: SLDL, Single-leg Lateral Drop Landing; CAI, Chronic Ankle Instability; CON, control.

## Discussion

The results of this study demonstrated that, during SLDL, the following angles were smaller in the CAI group than in the Coper group: (1) ankle dorsiflexion angle at 110–150 ms after landing, (2) maximum ankle dorsiflexion angle after landing, and (3) hip adduction angle at 160 ms after landing. Furthermore, compared to the CON group, the CAI group had smaller MH inversion/eversion excursions after landing. These results partially support the hypothesis of this study. No previous studies have reported comparing the biomechanics of the lower extremity of SLDL in the CAI, Coper, and CON groups; therefore, this study presents new data.

**Table 4. Hip kinematics data during 200 ms interval pre-landing.**

|  | CAI | Coper | CON | P-value | Effect Size |
|---|---|---|---|---|---|
| Flexion maximum (deg) | 31.6 ± 7.1 | 29.4 ± 8.0 | 30.8 ± 8.4 | .687 | .01 |
| Extension maximum (deg) | −19.7 ± 7.4 | −17.4 ± 5.5 | −18.1 ± 5.9 | .541 | .02 |
| Sagittal plane excursion (deg) | 11.9 ± 3.2 | 12.0 ± 4.0 | 12.7 ± 4.1 | .792 | .01 |
| Adduction maximum (deg) | −3.9 ± 3.8 | −2.5 ± 4.6 | −3.6 ± 5.7 | .634 | .02 |
| Abduction maximum (deg) | 17.8 ± 3.0 | 15.6 ± 3.6 | 17.1 ± 5.4 | .251 | .05 |
| Frontal plane excursion (deg) | 13.9 ± 2.7 | 13.1 ± 3.6 | 13.5 ± 3.2 | .749 | .01 |
| Internal rotation maximum (deg) | −13.8 ± 9.5 | −15.8 ± 8.4 | −13.2 ± 8.1 | .628 | .02 |
| External rotation maximum (deg) | 22.1 ± 10.1 | 23.1 ± 9.1 | 21.4 ± 8.4 | .850 | .01 |
| Horizonal plane excursion (deg) | 8.3 ± 4.6 | 7.3 ± 4.1 | 8.2 ± 3.2 | .686 | .01 |

Abbreviations: CAI, Chronic Ankle Instability; CON, control.

Mean ± SD

**Table 5. Hip kinematics data during 200 ms interval post-landing.**

| | CAI | Coper | CON | P-value | Effect Size |
|---|---|---|---|---|---|
| Flexion maximum (deg) | 34.9 (17.8) | 33.4 (16.1) | 36.0 (9.9) | .811 | .01 |
| Extension maximum (deg) | −20.1 ± 7.7 | −17.9 ± 5.4 | −18.4 ± 6.1 | .537 | .02 |
| Sagittal plane excursion (deg) | 17.5 (10.0) | 18.5 (10.5) | 16.7 (7.2) | .740 | .01 |
| Adduction maximum (deg) | −3.3 (4.4) | −0.3 (4.1) | −3.9 (9.3) | .078 | .07 |
| Abduction maximum (deg) | 17.1 ± 2.9 | 15.1 ± 3.3 | 16.6 ± 5.4 | .260 | .05 |
| Frontal plane excursion (deg) | 13.2 ± 3.3 | 13.8 ± 2.5 | 13.0 ± 3.4 | .678 | .01 |
| Internal rotation maximum (deg) | −13.2 ± 9.8 | −14.9 ± 8.3 | −13.5 ± 8.6 | .837 | .01 |
| External rotation maximum (deg) | 23.8 ± 10.2 | 23.4 ± 9.0 | 23.9 ± 9.3 | .980 | .00 |
| Horizonal plane excursion (deg) | 10.3 (7.1) | 7.4 (4.8) | 8.4 (6.3) | .287 | .05 |

Abbreviations: CAI, chronic ankle instability; CON, control.

Mean ± SD; Median (interquartile range)

Compared to the Coper group, the CAI group exhibited a smaller ankle dorsiflexion angle at 110–150 ms and a smaller maximum ankle dorsiflexion angle after landing (Fig 5, Table 3). This supports the hypothesis of this study. In a systematic review [11, 12] and several controlled studies [13–15, 27], CAI patients exhibited smaller ankle dorsiflexion angles during sports activities, including after jump landings. Previous case reports have documented that giving way during a forward drop jump landing on an inclined table exhibited smaller ankle dorsiflexion angles after landing with a giving way injury, compared to successful trials [33]. The results of this study support those observations. One of the symptoms of CAI is the lack of a dorsiflexion angle in the ankle joint during movement. This is caused by a limited ankle

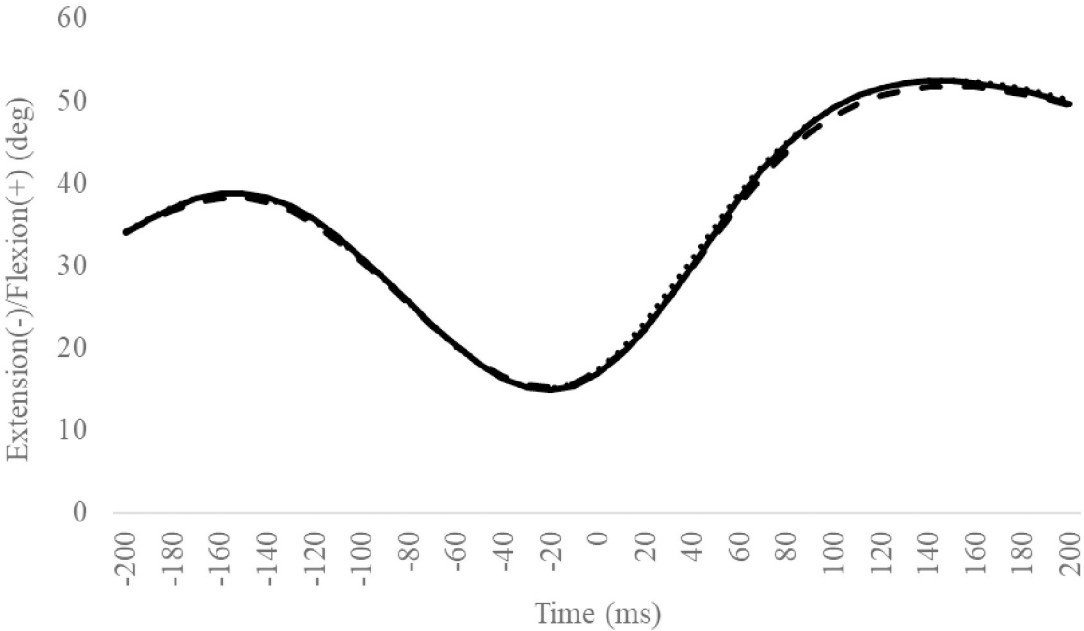

**Fig 11. Sagittal knee angle during SLDL from 200 ms before landing to 200 ms after landing.** The continuous, dotted, and dashed lines represent CAI, Coper, and CON, respectively. The time origin at 0 ms indicates the moment of initial contact. Abbreviations: SLDL, Single-leg Lateral Drop Landing; CAI, Chronic Ankle Instability; CON, control.

**Table 6. Knee kinematics data during 200 ms interval pre-landing.**

|  | CAI | Coper | CON | P-value | Effect Size |
|---|---|---|---|---|---|
| Flexion maximum (deg) | 40.1 ± 9.4 | 40.3 ± 8.2 | 40.1 ± 9.2 | .998 | .00 |
| Extension maximum (deg) | −12.1 (9.9) | −14.5 (8.0) | −14.8 (7.3) | .861 | .00 |
| Sagittal plane excursion (deg) | 25.5 ± 6.1 | 25.6 ± 6.7 | 25.5 ± 6.8 | .996 | .00 |

Abbreviations: CAI, Chronic Ankle Instability; CON, control.

Mean ± SD; Median (interquartile range)

**Table 7. Knee kinematics data during 200 ms interval post-landing.**

|  | CAI | Coper | CON | P-value | Effect Size |
|---|---|---|---|---|---|
| Flexion maximum (deg) | 53.4 ± 9.9 | 53.5 ± 5.8 | 52.5 ± 6.8 | .905 | .00 |
| Extension maximum (deg) | −13.5 (12.5) | −16.4 (9.2) | −16.1 (6.5) | .699 | .00 |
| Sagittal plane excursion (deg) | 36.6 ± 6.9 | 36.1 ± 6.3 | 35.4 ± 5.7 | .833 | .01 |

Abbreviations: CAI, Chronic Ankle Instability; CON, control.

Mean ± SD; Median (interquartile range)

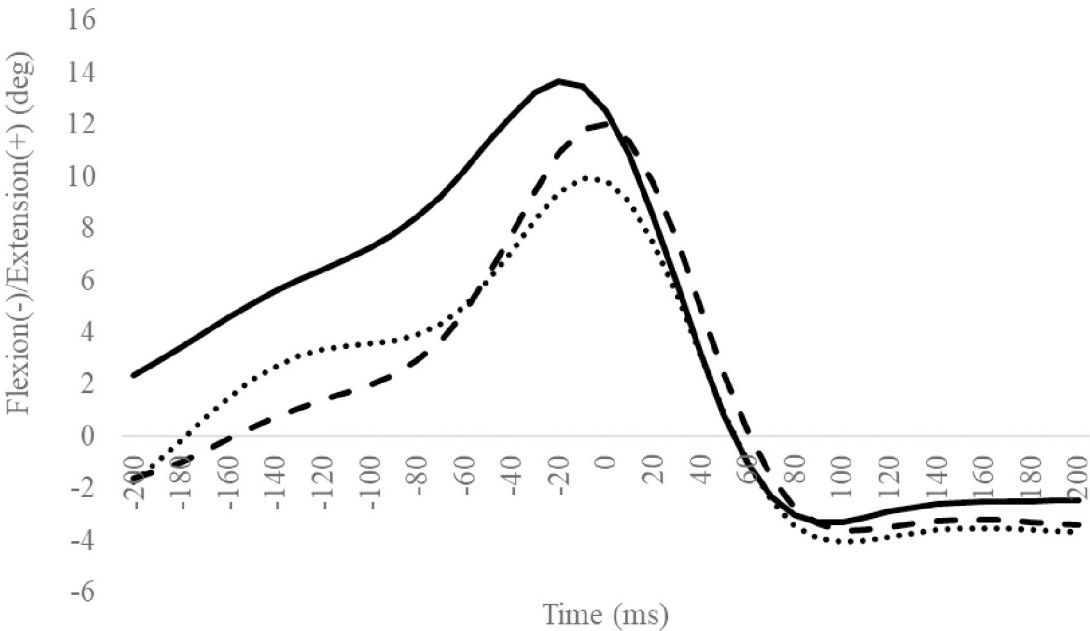

**Fig 12. Sagittal TM angle during SLDL from 200 ms before landing to 200 ms after landing.** The continuous, dotted, and dashed lines represent CAI, Coper, and CON, respectively. The time origin at 0 ms indicates the moment of initial contact. Abbreviations: SLDL, Single-leg Lateral Drop Landing; CAI, Chronic Ankle Instability; CON, control; TM, toe and metatarsal; MH, metatarsal and hind foot.

dorsiflexion range of motion and changes in movement patterns, which can easily result in ankle and subtalar inversion and adduction [34]. Therefore, an insufficient ankle dorsiflexion angle during movement is a predictor of LAS and CAI development [35, 36]. The differences in ankle dorsiflexion angles in this study occurred within the 100–200 ms interval during

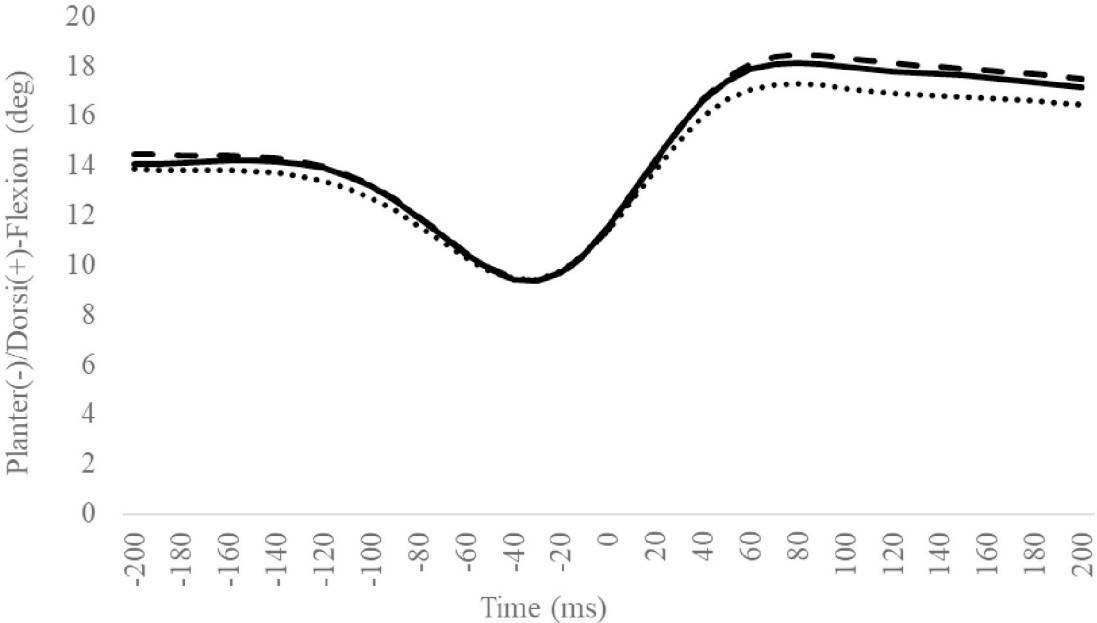

**Fig 13. Sagittal MH angle during SLDL from 200 ms before landing to 200 ms after landing.** The continuous, dotted, and dashed lines represent CAI, Coper, and CON, respectively. The time origin at 0 ms indicates the moment of initial contact. Abbreviations: SLDL, Single-leg Lateral Drop Landing; CAI, Chronic Ankle Instability; CON, control; TM, toe and metatarsal; MH, metatarsal and hind foot.

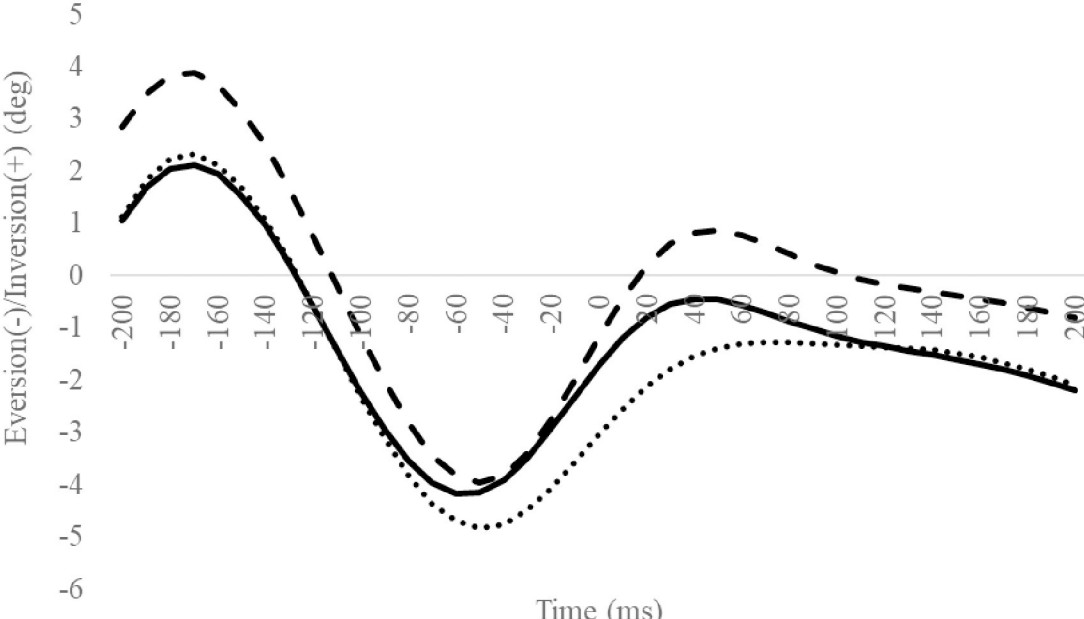

**Fig 14. Frontal MH angle during SLDL from 200 ms before landing to 200 ms after landing.** The continuous, dotted, and dashed lines represent CAI, Coper, and CON, respectively. The time origin at 0 ms indicates the moment of initial contact. Abbreviations: SLDL, Single-leg Lateral Drop Landing; CAI, Chronic Ankle Instability; CON, control; TM, toe and metatarsal; MH, metatarsal and hind foot.

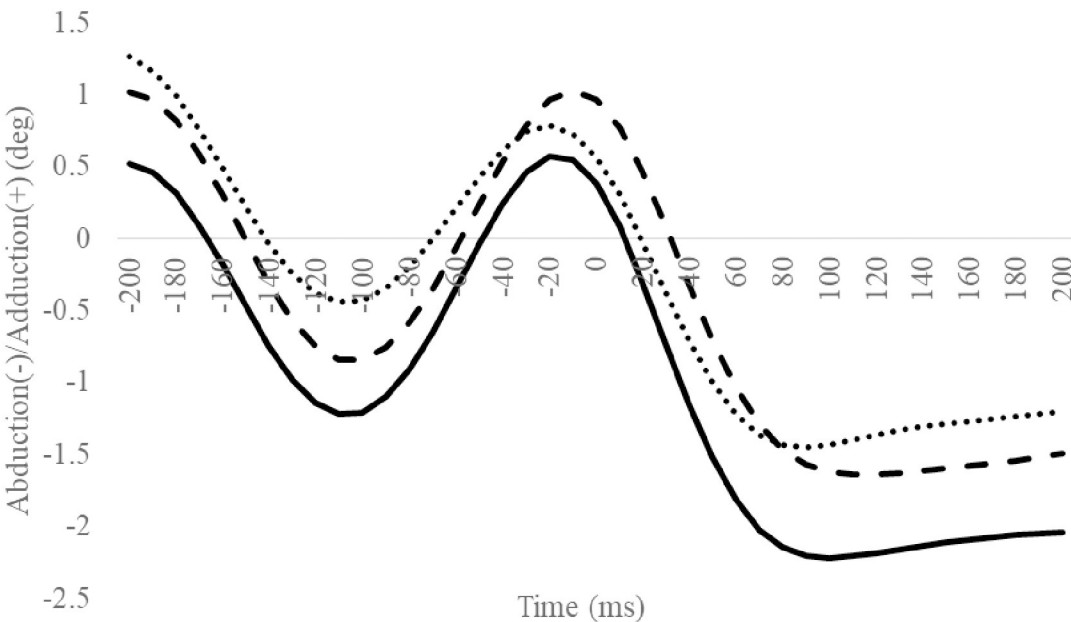

**Fig 15. Horizontal MH angle during SLDL from 200 ms before landing to 200 ms after landing.** The continuous, dotted, and dashed lines represent CAI, Coper, and CON, respectively. The time origin at 0 ms indicates the moment of initial contact. Abbreviations: SLDL, Single-leg Lateral Drop Landing; CAI, Chronic Ankle Instability; CON, control; TM, toe and metatarsal; MH, metatarsal and hind foot.

which LAS is more likely to occur [37]. Thus, compared to the Coper group, the CAI group may be more susceptible to LAS injuries during SLDL.

Compared to the Coper group, the CAI group exhibited a smaller hip adduction angle at 160 ms (Fig 8). Furthermore, compared to the CON group, the CAI group exhibited smaller MH inversion/eversion excursions after landing (Table 9). This observation does not support the hypothesis of this study and is a new finding. Case reports document that, compared to a

**Table 8. TM/MH kinematics data during 200 ms interval pre-landing.**

|  | CAI | Coper | CON | P-value | Effect Size |
|---|---|---|---|---|---|
| TM Flexion maximum (deg) | −1.5 ± 9.8 | 2.8 ± 8.9 | 2.2 ± 6.7 | .266 | .05 |
| TM Extension maximum (deg) | 14.8 ± 8.0 | 11.4 ± 7.0 | 11.9 ± 4.5 | .245 | .05 |
| Sagittal plane excursion (deg) | 13.4 ± 5.9 | 14.2 ± 4.2 | 14.0 ± 4.3 | .852 | .01 |
| MH Plantarflexion maximum (deg) | −9.3 ± 3.1 | −9.3 ± 3.2 | −9.3 ± 3.0 | .999 | .00 |
| MH Dorsiflexion maximum (deg) | 14.5 (3.4) | 14.7 (2.9) | 14.7 (3.5) | .826 | .01 |
| Sagittal plane excursion (deg) | 5.2 ± 1.6 | 4.7 ± 1.8 | 5.4 ± 1.6 | .505 | .03 |
| MH Inversion maximum (deg) | 3.2 (5.4) | 3.7 (8.6) | 5.5 (6.9) | .615 | .02 |
| MH Eversion maximum (deg) | 3.8 (4.5) | 5.6 (4.9) | 2.5 (7.3) | .371 | .01 |
| Frontal plane excursion (deg) | 7.2 ± 3.0 | 8.4 ± 3.6 | 9.0 ± 3.8 | .272 | .05 |
| MH Adduction maximum (deg) | 1.2 ± 2.5 | 1.8 ± 2.5 | 1.6 ± 2.0 | .743 | .01 |
| MH Abduction maximum (deg) | 1.4 ± 2.5 | 0.7 ± 2.5 | 1.0 ± 1.9 | .657 | .02 |
| Horizonal plane excursion (deg) | 2.6 ± 0.7 | 2.5 ± 0.7 | 2.6 ± 0.7 | .896 | .00 |

Abbreviations: CAI, Chronic Ankle Instability; CON, control; TM, toe and metatarsal; MH, metatarsal and hind foot.

Mean ± SD; Median (interquartile range)

**Table 9. TM/MH kinematics data during 200 ms interval post-landing.**

|  | CAI | Coper | CON | P-value | Effect Size |
|---|---|---|---|---|---|
| TM Flexion maximum (deg) | 3.6 ± 3.5 | 4.4 ± 3.4 | 4.0 ± 2.8 | .756 | .01 |
| TM Extension maximum (deg) | 12.5 ± 6.7 | 9.9 ± 5.7 | 12.2 ± 4.2 | .297 | .04 |
| Sagittal plane excursion (deg) | 16.1 ± 4.4 | 14.3 ± 3.9 | 16.3 ± 2.9 | .208 | .06 |
| MH Plantarflexion maximum (deg) | −11.6 ± 2.8 | −11.4 ± 3.0 | −11.4 ± 2.8 | .989 | .00 |
| MH Dorsiflexion maximum (deg) | 18.2 ± 2.1 | 17.4 ± 1.8 | 18.5 ± 2.7 | .273 | .05 |
| Sagittal plane excursion (deg) | 6.6 ± 2.5 | 6.0 ± 2.1 | 7.1 ± 2.1 | .299 | .04 |
| MH Inversion maximum (deg) | 0.4 (7.7) | 0.0 (7.1) | 2.8 (10.8) | .558 | .03 |
| MH Eversion maximum (deg) | 1.0 (5.1) | 3.8 (5.8) | 0.0 (10.0) | .408 | .01 |
| Frontal plane excursion (deg)* | 2.6 ± 1.0 | 3.0 ± 1.6 | 4.3 ± 2.4 | .027 | .15 |
| MH Adduction maximum (deg) | 0.4 ± 2.8 | 0.6 ± 2.5 | 1.0 ± 2.0 | .772 | .01 |
| MH Abduction maximum (deg) | 2.4 ± 2.7 | 1.6 ± 2.3 | 1.9 ± 2.1 | .548 | .02 |
| Horizonal plane excursion (deg) | 2.9 ± 1.3 | 2.2 ± 1.1 | 2.9 ± 1.4 | .180 | .06 |

Abbreviations: CAI, Chronic Ankle Instability; CON, control; TM, toe and metatarsal; MH, metatarsal and hind foot.

Mean ± SD; Median (interquartile range)

*CAI group and CON group differ significantly at p < .05

successful trial, giving way during a forward drop jump landing on an inclined table exhibited a smaller hip adduction angle after landing during the giving way kinematics [33]. The results of this study partially support this observation. Varying hip kinematics during movement is a predictor of CAI development [36]. The MH joint, consisting of the midfoot and hindfoot, plays vital role in regulating the flexibility and stiffness of the foot [38]. Low MH joint mobility increases foot stiffness [39, 40]. In this study, the amount of MH inversion/eversion excursion in the CAI group was smaller than in the CON group. This outcome may be ascribed to a rigid foot with reduced MH mobility [39, 40], a condition that makes it difficult to deform the foot flexibly, thus inhibiting its ability to absorb shocks efficiently, which may be considered a contributing factor to the risk of LAS. Therefore, compared to the Coper and CON groups, the CAI group may be more susceptible to LAS injuries during SLDL.

Two clinical implications can be identified from the results of this study. One, there is a need to assess SLDL. The results of this study revealed different kinematics, in contrast to previous reports that analyzed forward jump landings. Lateral jump landings cause tensile stress on the eversion muscle of the ankle and the lateral ligaments of the ankle joint [6, 17] and may influence the occurrence of LAS and the progression of CAI. In this study, we were able to elucidate this from a kinematic perspective. Two, this study suggests which specific joints must be focused on during SLDL. A detailed evaluation of hip adduction/abduction and plantar/dorsiflexion kinematics during SLDL is need.

This was the first study to analyze SLDL among the three designated groups. The results of this study highlight the need to assess single-leg drop jump landings, in CAI patients, not only in the forward direction but also to the side. In particular, a focus on the kinematics of hip adduction/abduction and plantar/dorsiflexion during the movement is suggested. However, this study had several limitations. First, the population of the study consisted of males aged 18–30 years with a Tegner Activity level of three or more. Therefore, it is unclear whether the results of this study may be generalized to other demographic groups. Second, because this was a cross-sectional study, the causal relationships in the study are unknown. Furthermore, it is unclear whether differences in kinematics during SLDL were caused by limitations in the range of motion of joints [35] or by changes in movement patterns resulting from central

nervous system adaptations [20]. Third, the amount of MH inversion/eversion excursion that was significant in this study was negligible, i.e., approximately 3˚, and may be difficult to evaluate in clinical practice. In future, the potential to propose preventive measures for CAI progression, may arise from intervention and prospective studies for elucidating the causal relationship between symptoms and biomechanics.

## Conclusions

The CAI group in this study exhibited:

- different kinematics of hip adduction/abduction, ankle plantar/dorsiflexion, and MH inversion/eversion during SLDL, when compared with the control and Coper groups, and

- a higher susceptibility to LAS injuries, which may be associated with smaller hip adduction and ankle dorsiflexion angles, as well as reduced motion in MH inversion/eversion after landing.

## Supporting information

**S1 File. In this file, kinematics data for the ankle, subtalar, hip, knee, TM, and MH are provided for each of 19 in the CAI group, 19 in the Coper group, and 19 in the CON group at intervals of 10 ms from -200 ms to 200 ms.**
(XLSX)

## Acknowledgments

We would like to thank Editage (www.editage.com) for English language editing.

## Author Contributions

**Conceptualization:** Yuki Sagawa, Takumi Yamada, Takehiro Ohmi, Yoshinao Moriyama, Junpei Kato.

**Data curation:** Takehiro Ohmi.

**Formal analysis:** Yuki Sagawa, Takumi Yamada, Takehiro Ohmi, Yoshinao Moriyama, Junpei Kato.

**Investigation:** Yuki Sagawa, Takumi Yamada, Takehiro Ohmi, Yoshinao Moriyama, Junpei Kato.

**Methodology:** Yuki Sagawa, Takumi Yamada, Takehiro Ohmi, Yoshinao Moriyama, Junpei Kato.

**Project administration:** Takehiro Ohmi.

**Supervision:** Takumi Yamada, Takehiro Ohmi, Yoshinao Moriyama, Junpei Kato.

**Validation:** Yuki Sagawa.

**Visualization:** Yuki Sagawa.

**Writing – original draft:** Yuki Sagawa.

**Writing – review & editing:** Yuki Sagawa.

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
