## [Decision Letter · Decision Letter 0]

24 Aug 2023

PONE-D-23-20707Differences in lower extremity kinematics during single-leg lateral drop landing of healthy subjects, copers, and subjects with chronic ankle instability.PLOS ONE

Dear Dr. sagawa,

Thank you for submitting your manuscript to PLOS ONE. After careful consideration, we feel that it has merit but does not fully meet PLOS ONE’s publication criteria as it currently stands. Therefore, we invite you to submit a revised version of the manuscript that addresses the points raised during the review process.

We look forward to receiving your revised manuscript.

Kind regards,

Kentaro Amaha

Academic Editor

PLOS ONE

Journal Requirements:

Additional Editor Comments (if provided):

Please address the issues identified by the reviewers appropriately.

Reviewers' comments:

Reviewer's Responses to Questions

**Comments to the Author**

1. Is the manuscript technically sound, and do the data support the conclusions?

Reviewer #1: Partly

Reviewer #2: Yes

2. Has the statistical analysis been performed appropriately and rigorously? 

Reviewer #1: I Don't Know

Reviewer #2: Yes

3. Have the authors made all data underlying the findings in their manuscript fully available?

Reviewer #1: Yes

Reviewer #2: Yes

4. Is the manuscript presented in an intelligible fashion and written in standard English?

Reviewer #1: Yes

Reviewer #2: Yes

5. Review Comments to the Author

Reviewer #1: First, I would like to express sincere gratitude to get an opportunity to review the manuscript. The endeavor of the authors is appreciated. However, there is some scope for its improvement.

Overall comments:

1. Kindly follow the guidelines to authors for structuring the manuscript.

2. Involvement of 3 groups and design mentioned as cross-sectional study do not sound convincing. Kindly explain.

3. The paper needs to be structured scientifically. For few examples, introduction is lengthy and method section contains information suitable for results section. Kindly do the needful.

4. It would be better if study design were clearly mentioned in title, abstract and main text.

5. It would be better to define primary and secondary outcome variables in methods section in line with objectives. Results, discussion and conclusion need to follow the same in chain.

6. Was informed written consent taken from the participants of the study?

Section wise comments

1. Please DELETE INFORMATION UNRELATED TO OBJECTIVE so that the section is short and sweet. Kindly shorten this section and delete unrelated information. Kindly focus on three elements of introduction

a. What is known about the topic? (Background)

b. What is not known? (The research problem)

c. Why the study was done? (Justification)

2. Methods sections is supposed to be core of any study. Here, methods section contains inadequate information. For example, following components for methods section need to be well described.

i. Details of Study design

ii. Setting

iii. Sample size estimation

iv. Sampling technique

v. Participant

vi. Primary and secondary outcome variables with working definition

vii. Intervention/issue of interest (exposure)

viii. Comparison

ix. Ethics and end point

x. Statistical analysis

3. Results

a. It would be better to start this section with baseline information of the patients.

b. The section needs to be placed in proper order such that primary outcome variables are mentioned first.

c. This section needs to be specific as per the objectives.

4. The discussion section needs to be described scientifically. Kindly frame it along the following lines:

a. Main findings of the present study

b. Comparison with other studies

c. Implication and explanation of findings

d. Strengths and limitations

e. Conclusion, recommendation and future directions

5. Conclusion needs to provide answers for each objective clearly in a sentence or two.

Reviewer #2: Overall Consideration:

In the manuscript, “Differences in lower extremity kinematics during single-leg lateral drop landing of healthy subjects, copers, and subjects with chronic ankle instability”, the authors utilized three-dimensional motion analyses to determine if ankle instability associated with different body kinematics. This is a cross-sectional, observational study. Prior studies have examined this topic, but did not utilize the lateral drop motion examined within this study. As cutting movements simulated by the lateral drop motion frequently lead to ankle injuries, this study will provide valuable information to the field. This is the main strength of the manuscript; however, it may be more suitable for a specialized journal as opposed to PLOS One (main criticism). A point-by-point review is below.

Major comments:

1. Can the authors comment on how their study could either impact preventative measures or clinical impact. They hint towards this, but the meaning of the study is lost within the discussion.

2. Comments on muscle strength and ankle instability could be added to the introduction.

3. The authors may wish to add a flow diagram depicting participant recruitment.

4. Line 419: The authors observe that hip adduction and ankle dorsiflexion angles, as well as MH inversion/eversion excursion after landing were greater in the chronic instability group; however, these two things are simply associated with each other within the study. The authors do not have causative data. They should temper this sentence and others like it.

6. PLOS authors have the option to publish the peer review history of their article (what does this mean?). If published, this will include your full peer review and any attached files.

Reviewer #1: **Yes: **Dr Satish Prasad Barnawal

Reviewer #2: No

---

## [Author Response · Author response to Decision Letter 0]

10 Nov 2023

Editor Comments

Response:　

Thank you for taking the time to review our paper. The following changes have been made in compliance with the style requirements of PLOS ONE.

Zip codes have been deleted from the affiliations. (Page 1, Line 7-14)

“Division of Physical Therapy, Department of Rehabilitation, ~” has been changed to “Department of Rehabilitation, Division of Physical Therapy, ~.” (Page 1, Line 12)

The word count of the Abstract is now less than three hundred words long. (Page 2, Abstract, Line 21-40)

Reference [33] is now provided as a URL in the text. (Page 7, Setting, Line 130)

Figures 1-8 have been sent as tiff files. (Pages 7-9, 12, 15, 17, 18)

2. In your Data Availability statement, you have not specified where the minimal data set underlying the results described in your manuscript can be found. 

Response:

An Excel, comprising the minimal data set (of each joint angle) underlying our results, has been attached.

 

Reviewer 1

Comment 1: Kindly follow the guidelines to authors for structuring the manuscript.

Response:　

Thank you for taking the time to review our paper. The following changes have been made in compliance with the style requirements provided by the journal’s editor.

Zip codes have been deleted from the affiliations. (Page 1, Line 7-14)

“Division of Physical Therapy, Department of Rehabilitation, ~” has been changed to “Department of Rehabilitation, Division of Physical Therapy, ~.” (Page 1, Line 12)

The abstract word count is now less than three hundred words long. (Page 2, Abstract, Lines 21-40)

Reference [33] is now provided as a URL in the text. (Page 7, Setting, Line 130)

Figures 1-8 have been sent as tiff files. (Pages 7-9, 12, 15, 17, 18)

Comment 2: Involvement of 3 groups and design mentioned as cross-sectional study do not sound convincing. Kindly explain.

Response:

Thank you for pointing this out.

2.(1) Reason for using three groups instead of two for a cross-sectional study.

We believe it is inappropriate to use only two groups, that is, the CAI group and the other groups (healthy + coper group). We made this choice because previous studies, such as those by S. JUN SON in 2017 and Seunguk Han in 2021, have already established that differences exist in the kinematics and kinetics during jump landing and cutting movements between healthy individuals and copers. Furthermore, it is important to consider that the presence of a history of ankle sprains and their associated symptoms may introduce confounding factors during jump landing assessments. In light of these considerations, we adopted a research design comprising three distinct groups rather than just two. This detailed explanation has been incorporated into the "Design" section of the Materials and Methods. (Page 5, Design, Lines 83-85). 

2.(2) Reason for the cross-sectional study design

From our perspective, before initiating the necessary longitudinal studies, it is important to first examine whether there are differences in kinematics among the three groups in a cross-sectional study.

Comment 3: The paper needs to be structured scientifically. For few examples, introduction is lengthy and method section contains information suitable for results section. Kindly do the needful.

Response:

Thank you for your advice. We have removed the unnecessary sentences from the Introduction and shortened it (Pages 3-5, Introduction, Lines 42-79), and moved Table 1 from the Methods section to the Results section (Pages 11-12, Results, Lines 205-213).

Comment 4: It would be better if study design were clearly mentioned in title, abstract and main text.

Response:

The title, abstract, and main text clearly state that the study has been designed as a cross-sectional observational study (Page 1, Title, Line 3; Page 2, Abstract, Line 27; Page 5, Design, Lines 82-85).

Comment 5: It would be better to define primary and secondary outcome variables in methods section in line with objectives. Results, discussion and conclusion need to follow the same in chain.

Response:

Thank you for your advice. We have defined the primary and secondary outcome variables in a new section, “Primary and secondary outcome variables”, under Methods (Page 10, Primary and secondary outcome variables, Lines 181-183).

Results, Discussion, and Conclusions follow in the same chain.

Comment 6: Was informed written consent taken from the participants of the study?

Response:

Thank you for your question. Yes, informed consent was obtained from the research participants as indicated in the section on Ethics (Page 11, Ethics, Lines 197-202).

Introduction

Comment 1:

Please delete information unrelated to objective so that the section is short and sweet. Kindly shorten this section and delete unrelated information. Kindly focus on three elements of introduction

a. What is known about the topic? (Background)

b. What is not known? (The research problem)

c. Why the study was done? (Justification)

Response:

Thank you for this advice. We have deleted the information unrelated to the objective and have simplified the background. Only items that are the basis for the hypothesis are now listed (Pages 3-5, Introduction, Lines 42-79). 

Methods

Comment 1:

Methods sections is supposed to be core of any study. Here, methods section contains inadequate information. For example, following components for methods section need to be well described.

i. Details of Study design

ii. Setting

iii. Sample size estimation

iv. Sampling technique

v. Participant

vi. Primary and secondary outcome variables with working definition

vii. Intervention/issue of interest (exposure)

viii. Comparison

ix. Ethics and end point

Response:

Thank you for your advice and the specific example. Please find the description of the revised components of the Methods section below.

i. The Design section provides the rationale for the three groups design of the cross-sectional observational study (Page 5, Design, Lines 83-85).

ii. The heading “Instrumentation” has been changed to "Setting". (Page 7-8, Setting, Lines 126-137)

iii. A “Sample size estimation” section that details the data and the statistical power analysis tool used has been added. (Page 5, Sample size estimation, Lines 87-92).

iv. A "Sampling technique" section has been added with information that convenience sampling was used (Page 5, Sampling technique, Lines 94-95).

v. Modifications have been made to the inclusion/exclusion criteria in the Participants section (Pages 6-7, Participants, Lines 97-122).

vi. A Primary and secondary outcome variables section has been added (Page 10, Primary and secondary outcome variables, Lines 181-183).

vii. The SLDL method, failure trial descriptions, and some citation numbers have been modified in the Procedures section (Page 8, Procedure, Line 143-156).

viii. The Introduction compares the three groups, the Design subsection justifies the inclusion of three groups in the cross-sectional study, and the Participants section provides the group inclusion criteria (Pages 3-4, Introduction, Lines 42-79; Page 5, Design, Lines 82-85; Pages 6-7, Participants, Lines 97-122).

ix. Ethical considerations, concerning informed consent and ethics committee approval, have been provided in the Ethics section. This study does not have endpoints because it is a cross-sectional study (Page 11, Ethics, Lines 197-202).

x. It has been specified that either One-way ANOVA or the Kruskal–Wallis test was used for statistical analysis, and that either the Tukey or Steel–Dwass test was used for the post-test (Pages 10-11, Statistical analysis, Lines 185-195). We have also deleted some content (Page 10-11, Statistical analysis, Lines 185-195) and created a new section called Sample size estimation (Page 5, Sample size estimation, Lines 87-92).

Results

Comment 1:

a. It would be better to start this section with baseline information of the patients.

b. The section needs to be placed in proper order such that primary outcome variables are mentioned first.

c. This section needs to be specific as per the objectives.

Response:

Thank you for your advice. We have made the following corrections:

a. The results now start with the patient's baseline (demographic) information (Pages 11-12, Results, Lines 204-213).

b. The primary outcome variables, ankle/subtalar kinematics, are mentioned first (Pages 12-14, Ankle/Subtalar angle, Lines 215-238).

c. The Knee Kinematics section was revised to "For all the outcomes, the CAI group showed no significant differences (Fig 7, Tables 6 and 7) compared to the Coper and CON groups". (Pages 17, Knee angle, Lines 262-278)

Discussion

Comment 1:

The discussion section needs to be described scientifically. Kindly frame it along the following lines:

a. Main findings of the present study

b. Comparison with other studies

c. Implication and explanation of findings

d. Strengths and limitations

e. Conclusion, recommendation and future directions

Response:

Thank you for your advice. We have revised the items you pointed out as follows.

Paragraph 1 has been revised to begin with the primary outcome results (Page 21, Discussion, Line 305-311).

In paragraph 2, comparisons of the primary outcome with other studies have been included (Pages 21-22, Discussion, Lines 312-326).

In paragraph 3, four paragraphs have been deleted. The secondary outcome and studies on the hip and foot have been combined in the same paragraph (Pages 22-23, Discussion, Lines 327-342).

In paragraph 4, we have described the clinical application of the content of the previous 6 paragraphs. The contents include the necessity of evaluating lateral jump landings and information on which joints to focus on when performing lateral jump landings (Page 23, Discussion, Lines 343-350).

The Limitation section was deleted and merged with paragraph 5. Corrections and additions were made to the strengths, limitations and future prospects (Pages 23-24, Discussion, Line 354-363). 

A Conclusions section has been added (Page 24, Conclusions, Line 367-374).

Conclusion

Comment 1: Conclusion needs to provide answers for each objective clearly in a sentence or two.

Response:

Thank you for this advice. We have deleted the previous sentence and replaced it with two bullet points. (Page 24, Conclusions, Lines 367-374)

 

Reviewer 2

In the manuscript, “Differences in lower extremity kinematics during single-leg lateral drop landing of healthy subjects, copers, and subjects with chronic ankle instability”, the authors utilized three-dimensional motion analyses to determine if ankle instability associated with different body kinematics. This is a cross-sectional, observational study. Prior studies have examined this topic, but did not utilize the lateral drop motion examined within this study. As cutting movements simulated by the lateral drop motion frequently lead to ankle injuries, this study will provide valuable information to the field. This is the main strength of the manuscript; however, it may be more suitable for a specialized journal as opposed to PLOS One (main criticism). A point-by-point review is below.

1. Can the authors comment on how their study could either impact preventative measures or clinical impact. They hint towards this, but the meaning of the study is lost within the discussion.

Response:

Thank you for your question. We do not believe that we can propose preventive measures since this is a cross-sectional study. This limitation is duly acknowledged in the fifth paragraph of our Discussion section (Pages 23-24, Discussion, Line 354-363). Nonetheless, our study yields two noteworthy clinical implications. Firstly, there is a pressing need to assess lateral jump landings as the kinematics we observed differ from previous studies that primary focused on anterior jump landings. Secondly, our findings suggest a specific focus on particular joints when evaluating lateral jump landings. In light of our results, we recognize the significance of conducting detailed assessments of hip adduction and plantar dorsiflexion kinematics during lateral jump landings. Two, the study suggests which joints to focus on when evaluating lateral jump landings. Based on the results of this study, we recognize that it will be important to evaluate the kinematics of hip adduction and plantar dorsiflexion in detail during lateral jump landings. We believe that longitudinal and interventional studies will be conducted in the future to clarify the causal relationship between the symptoms and kinematics, which will facilitate preventive measures and further clinical adaptations.

2. Comments on muscle strength and ankle instability could be added to the introduction.

Response:

Thank you for your advice. The purpose of this study is to differentiate kinematics, and muscle strength; furthermore, kinetics are not raised in the hypothesis. Of course, the effect of muscle weakness after injury may exist; however, we did not add it to the introduction because it is not the purpose of this study.

3. The authors may wish to add a flow diagram depicting participant recruitment.

Response:

Thank you for your advice. We have created a flow diagram (Figure 1) and added it to the Methods section. (Page 7, Participants, Line 124)

4. Line 419: The authors observe that hip adduction and ankle dorsiflexion angles, as well as MH inversion/eversion excursion after landing were greater in the chronic instability group; however, these two things are simply associated with each other within the study. The authors do not have causative data. They should temper this sentence and others like it.

Response:

In this study, the Chronic Ankle Instability (CAI) group displayed:

1. Distinct kinematic patterns in terms of hip adduction/abduction, ankle plantar/dorsiflexion, and the joints between the metatarsals and the hindfoot (the MH joint) inversion/eversion during Single-leg Lateral Drop Landing (SLDL) when compared to the control and Coper groups.

2. A heightened vulnerability to lateral ankle sprains (LAS), which could be linked to smaller hip adduction and ankle dorsiflexion angles, as well as reduced motion in MH inversion/eversion following landing.” (Page 24, Conclusions, Lines 367-374).

---

## [Decision Letter · Decision Letter 1]

10 Jan 2024

Differences in lower extremity kinematics during single-leg lateral drop landing of healthy individuals, injured but asymptomatic patients, and patients with chronic ankle instability- a cross-sectional observational study.

PONE-D-23-20707R1

Dear Dr. sagawa,

We’re pleased to inform you that your manuscript has been judged scientifically suitable for publication and will be formally accepted for publication once it meets all outstanding technical requirements.

Kind regards,

Kentaro Amaha

Academic Editor

PLOS ONE

Additional Editor Comments (optional):

I believe that many of the corrections have been well addressed and are worthy of publication.

Reviewers' comments:

Reviewer's Responses to Questions

**Comments to the Author**

1. If the authors have adequately addressed your comments raised in a previous round of review and you feel that this manuscript is now acceptable for publication, you may indicate that here to bypass the “Comments to the Author” section, enter your conflict of interest statement in the “Confidential to Editor” section, and submit your "Accept" recommendation.

Reviewer #1: (No Response)

2. Is the manuscript technically sound, and do the data support the conclusions?

Reviewer #1: Partly

3. Has the statistical analysis been performed appropriately and rigorously? 

Reviewer #1: I Don't Know

4. Have the authors made all data underlying the findings in their manuscript fully available?

Reviewer #1: Yes

5. Is the manuscript presented in an intelligible fashion and written in standard English?

Reviewer #1: Yes

6. Review Comments to the Author

Reviewer #1: Congratulations to the authors for the study. It can have significant contributions to the scientific community.

7. PLOS authors have the option to publish the peer review history of their article (what does this mean?). If published, this will include your full peer review and any attached files.

Reviewer #1: No

---

## [Editor Report · Acceptance letter]

12 Mar 2024

PONE-D-23-20707R1 

PLOS ONE

Dear Dr. Sagawa, 

I'm pleased to inform you that your manuscript has been deemed suitable for publication in PLOS ONE. Congratulations! Your manuscript is now being handed over to our production team.

Kind regards, 

on behalf of

Dr. Kentaro Amaha 

Academic Editor

PLOS ONE